# CLIP meets Model Zoo Experts: Pseudo-Supervision for Visual Enhancement

## Abstract

Contrastive language image pretraining (CLIP) is a standard method for training vision-language models. While CLIP is scalable, promptable, and robust to distribution shifts on image classification tasks, it lacks object localization capabilities. This paper studies the following question: *Can we augment CLIP training with task-specific vision models from model zoos to improve its visual representations?* Towards this end, we leverage open-source task-specific vision models to generate pseudo-labels for an uncurated and noisy image-text dataset. Subsequently, we train CLIP models on these pseudo-labels in addition to the contrastive training on image and text pairs. This simple setup shows substantial improvements of up to 16.3% across different vision tasks, including segmentation, detection, depth estimation, and surface normal estimation. Importantly, these enhancements are achieved without compromising CLIP's existing capabilities, including its proficiency in promptable zero-shot classification.

## 1 Introduction

Foundation Models (FMs) are revolutionizing different domains of artificial intelligence and machine learning, including computer vision (Radford et al., 2021; He et al., 2022; Kirillov et al., 2023b) and natural language processing (Devlin et al., 2018; Brown et al., 2020; Touvron et al., 2023). FMs can be trained on web crawled data without relying on crowd or expert annotations, and yet they demonstrate strong generalization capabilities (Jia et al., 2021; Schuhmann et al., 2022).

CLIP, one of the most prominent methods for FM training in vision, uses contrastive learning to align image and text representations (Radford et al., 2021; Jia et al., 2021). In addition to robustness to data distribution shifts, CLIP offers impressive zero-shot and cross-modal retrieval capabilities on unseen datasets. Nevertheless, computer vision encompasses a broad range of tasks that require the ability to comprehend spatial relationships, semantic content, object localization, and 3D structures. In spite of CLIP's impressive zero-shot open-vocabulary classification accuracy, it exhibits poor localization capabilities and often struggles in associating text with objects in an image (Thrush et al., 2022; Ghiasi et al., 2022; Ranasinghe et al., 2023). Consequently, in practice, many vision tasks (e.g., detection and segmentation), rely on CLIP through fine-tuning the entire model to compensate for these localization deficiencies.

In this work, we seek to answer the following question: *Can we augment CLIP training with task-specific vision models from model zoos to improve its visual representations?* That is, we seek to (1) use open-source task-specific vision models to generate *hard* pseudo-labels on a web-scale noisy image-text dataset and (2) train CLIP on image-text pairs along with pseudo-labels with multiple objectives. An overview of our approach, which we call **CLIP T**raining with **eX**perts (**CLIPTeX**), is shown in Fig. 1. We show that **CLIPTeX** enhances the visual representations of CLIP while retaining the pre-existing capabilities of CLIP.

We summarize our contributions as follows:

- We introduce simple and effective method, **CLIPTeX**, to improve the visual representations of CLIP by leveraging experts specialized in object localization, depth estimation, and surface normal estimation. Through the generation of *hard* pseudo-labels on a noisy image-text dataset and the training of CLIP on these paired data points with multiple objectives, we achieve a significant

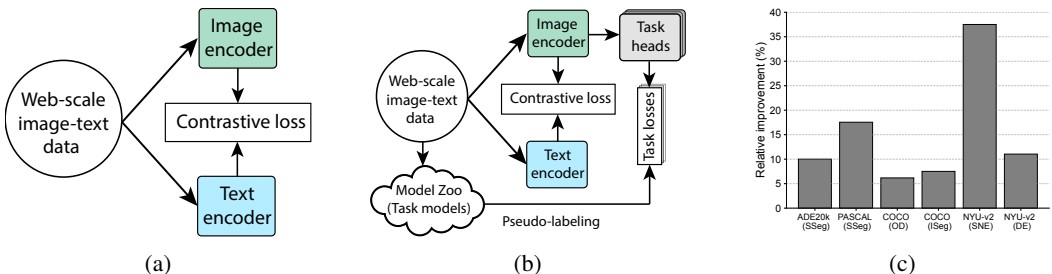

Figure 1: **Training CLIP with pseudo-labels improves its visual representations.** (a) shows the standard CLIP training. (b) shows **CLIPTeX** that trains CLIP with pseudo-labels from experts. Note that the main purpose of task heads is to improve CLIP's image encoder with expert knowledge, and the heads can be discarded after training. (c) shows the relative improvement that **CLIPTeX** obtains over CLIP-FT. Note that CLIP-FT is a stronger baseline than CLIP (see Section 4 and Section 5 for details). Here, SSeg, OD, ISeg, SNE, and DE refer to semantic segmentation, object detection, instance segmentation, surface normal estimation, and depth estimation respectively.

improvement in visual representations. Notably, our method yields up to 16.3% enhancement in probing accuracy across a diverse set of vision tasks and datasets.

- Our approach leads to positive transfer of representations to down-stream tasks and preserves the inherent strengths of CLIP, including its ability to perform zero-shot classification. This ensures that the model remains versatile and applicable across a wide range of computer vision domains.

- Experiments with multiple probes on variety of vision tasks and datasets (e.g., segmentation on PASCAL VOC and ADE20k, detection on COCO, depth estimation on NYU-v2, classification on ImageNet-1k and Places-365, and surface normal estimation on NYU-v2) demonstrate the effectiveness of **CLIPTeX** .

## 2 RELATED WORK

**Vision FMs.** Vision FMs extended the concept of pre-training to vast datasets containing hundreds of millions or even billions of images. This was in part driven by the introduction of ViTs (Dosovitskiy et al., 2020) which demonstrated the scalability of training Transformers (Vaswani et al., 2023) to such large-scale datasets in the field of computer vision. Since then, numerous large-scale pre-training methods have emerged in the domain of computer vision (e.g., Radford et al., 2021; Yu et al., 2022; Caron et al., 2021; He et al., 2022). Arguably, one of the most prominent classes of vision FMs is CLIP that specializes in aligning noisy image-text pairs from the web (Radford et al., 2021; Schuhmann et al., 2022; Gadre et al., 2023). This distinction is not only attributed to its scalability, but also to its prompting capabilities and robustness in handling dataset distribution shifts. Nevertheless, these models often face challenges in associating text with individual objects and localizing them (Thrush et al., 2022; Ghiasi et al., 2022; Ranasinghe et al., 2023). This work focuses on enhancing this capability through pseudo-supervision.

**Pseudo-supervision with experts.** The primary objective of pseudo-supervision (Lee et al., 2013) is to facilitate model training by generating pseudo-labels for unlabeled data, typically leveraging experts trained on a subset of the data containing ground truth labels. This methodology has also been applied to the training of foundation models (FMs). To the best of our knowledge, current approaches involve the acquisition of crowd labels for a portion of the data on a *single* task, with the subsequent training of experts on this labeled subset (e.g., Ghiasi et al., 2021; Zhang et al., 2022; Kirillov et al., 2023a; Liu et al., 2023). These trained experts are then utilized to create pseudo-labels for the remaining unlabeled data. Essentially, these methods employ experts that have been trained on the same or similar data distribution as the unlabeled data, aiming to achieve positive transfer. For example, in GLIP (Li et al., 2022a), a subset of web data is crowd-sourced to obtain localization labels, which is then used for expert training. Following expert training, these experts are employed to generate pseudo-labels for the remaining unlabeled web data. This combination of crowd labels and pseudo-labels is subsequently used to train the GLIP V2 (Zhang et al., 2022) model. SAM

(Kirillov et al., 2023a) also follows similar paradigm for creating large-scale segmentation dataset. Unlike previous approaches, our proposed method uses publicly accessible experts trained on diverse tasks with different data distributions and objectives.

**Multi-task learning for FMs.** Multi-tasking (Caruana, 1997; Ruder, 2017), a standard method for training on multiple tasks simultaneously, is widely used in machine learning (Liu et al., 2019; Misra et al., 2016; Sun et al., 2020), including FMs (e.g., Wang et al., 2022; Chen et al., 2022; Yang et al., 2021; Shukor et al., 2023; Zhang et al., 2023). Existing multi-task FMs creates a unified multi-task datasets by either collecting a new labeled dataset (e.g., Sun et al., 2022) or mixing existing labeled datasets (e.g., Lu et al., 2022), to facilitate positive transfer of knowledge to down-stream tasks. In contrast, **CLIPTeX** does not need any data collection and uses pseudo-supervision for training.

## 3 **CLIPTeX**

CLIP, a scalable FM, aligns image and text representations obtained from independent image and text encoders using contrastive loss. Although it offers zero-shot and cross-modal retrieval capabilities, it exhibits poor localization and dense prediction capabilities. This work, **CLIPTeX**, extends CLIP with pseudo-supervision from publicly available task experts specializing in localization, depth, and surface normal estimation. Our approach enhances CLIP's representations *without any labeled data collection* (Fig. 1).

### 3.1 MODELING

**Image and text encoders in CLIPTeX.** To train on image-text datasets, **CLIPTeX** uses two encoders, similar to CLIP: (1) an image encoder that takes RGB image as an input and produces an image embedding as an output and (2) a text encoder, that takes the text caption as an input and produces a text embedding as its output. Contrastive loss $\mathcal{L}_{\text{clip}}$ between image and text embeddings is one of the losses used to train **CLIPTeX**, as in CLIP.

**Task-specific heads.** To train **CLIPTeX** with pseudo-labels, we use task-specific heads that take the output of image encoder as an input and generate predictions for the respective task. Previous work have shown that multi-scale representations provides significant benefit in tasks requiring localization and fine-grained visual understanding (Zhao et al., 2017; Lin et al., 2017). However, some image encoders (e.g., ViT) do not inherently possess these capabilities. To ensure **CLIPTeX** can learn better visual representations independent of the image backbone used, we include a single shared multi-scale module (Zhao et al., 2017) between image encoder and task-specific heads. We feed the output of the image encoder through a multi-scale module (Zhao et al., 2017), which in turn feeds into the lightweight task-specific classification or regression heads. In our implementation, we use independent point-wise convolution as the head for each task.

Note that the spatial dimensions of the output from the task head may not be the same as the input image resolution, and certain tasks (e.g., segmentation, depth and surface normal estimation) may require them to be similar. In such instances, we perform nearest neighbour interpolation on head's output.

### 3.2 TRAINING

To train **CLIPTeX** with pseudo-supervision on $n$ tasks, we first generate *hard* pseudo-labels offline using publicly available task-specific experts. This is done on an uncurated web-scale dataset. We then train **CLIPTeX** with a weighted sum of contrastive loss in CLIP $\mathcal{L}_{\text{clip}}$ and task-specific losses $\mathcal{L}_{\text{task}}$, as:

$$\mathcal{L} = \lambda_{\text{clip}} \cdot \mathcal{L}_{\text{clip}} + \sum_{t=1}^{n} \lambda_{\text{task}}^{t} \cdot \mathcal{L}_{\text{task}}^{t}, \tag{1}$$

where $\mathcal{L}_{\text{task}}^{t}$ is the loss of the $t$-th task. Here, $\lambda_{\text{task}}^{t}$ and $\lambda_{\text{clip}}$ are the loss coefficients of $t$-th task and the standard CLIP loss, respectively.

# 4 EXPERIMENTAL SETUP

Probing is a standard method to study the representations learnt by neural networks (Alain & Bengio, 2016; He et al., 2022; Radford et al., 2021). We probe **CLIPTeX** and other pre-trained models on different down-stream tasks and multiple datasets using *classifier* or *regressor probes*. This helps us understand if training with hard pseudo-labels from experts can improve the effectiveness of CLIP's image representations across different vision tasks. In the following sub-sections, we provide details of our experimental setup, including expert models for pseudo-labeling (Section 4.1), baseline methods (Section 4.2), probes (Section 4.3), and down-stream tasks and datasets that are used for probing (Section 4.4).

## 4.1 TASK-SPECIFIC EXPERTS

Images provide valuable visual information about the appearance of objects in a scene, such as texture, color, and shape. However, they lack information about the spatial relationships between objects, specifically their relative distances and orientation. To aid CLIP's image encoder in learning more comprehensive representations of visual content within an image, including object's orientation and location, we generate hard pseudo-labels from the following publicly available experts:

- **Semantic segmentation.** We use Mask-RCNN (He et al., 2017) with ViT backbone (Dosovitskiy et al., 2020), trained on the COCO (Lin et al., 2014) with RangeAugment (Mehta et al., 2022b), to produce pseudo-labels for segmentation.

- **Monocular depth estimation:** We use DPT (Ranftl et al., 2021), trained on MIX-6 dataset (Ranftl et al., 2021), to generate monocular depth map pseudo-labels.

- **Surface normal estimation:** We use *NLL-AngMF* (Bae et al., 2021) as our surface normal expert, which is trained on ScanNet dataset (Dai et al., 2017).

## 4.2 CLIP BASELINES

CLIP models pre-trained on billions of images have demonstrated impressive generalization properties. However, training on such scale datasets with experts at high resolution (e.g., input image is $512 \times 512$)[1] is computationally expensive. Therefore, to show the efficacy of our approach, we finetune pre-trained CLIP with and without pseudo-labels on CC3M (Sharma et al., 2018). We compare with the following baselines to show the efficacy of pseudo-supervision:

- **CLIP.** We use CLIP model by Mehta et al. (2022a) pretrained on 1.2 billion images with a variable resolution and batch sampler whose base input image's spatial resolution is $224 \times 224$. The model is robust to input image's scale (likely due to multi-resolution training) and also, more performant compared to other open-source CLIP models trained on similar dataset size (e.g., OpenCLIP of Ilharco et al. (2021)).

- **CLIP-FT.** Many dense prediction tasks (e.g., segmentation) exhibit enhanced performance when provided with high-resolution input images. However, CLIP is initially pre-trained on images with $224 \times 224$ spatial resolution. Therefore, to gain better insights into the benefits of pseudo-supervision, we finetune CLIP with contrastive loss on CC3M's image and text pairs. For this fine-tuning, we employ variable resolution and a batch sampler, whose base input image's spatial resolution is $512 \times 512$. Importantly, this model serves as a fairer baseline for **CLIPTeX** compared to CLIP, as it has been adapted to CC3M data with high-resolution images. Therefore, any improvements observed over this baseline signify a pure transfer of knowledge from pseudo-supervision.

  To show the generality of our approach, we conducted experiments with three image encoder backbones: ViT-B/16, ViT-H/16, and ResNet-50. Also note that we finetune **CLIPTeX** on CC3M's image and text pairs along with pseudo-labels (Section 4.1) using the same settings as CLIP-FT. We use cross-entropy loss to train on segmentation pseudo-labels, and L1 loss to train on depth and surface normal pseudo-labels.

---

[1]Many dense prediction tasks (e.g., segmentation and detection) require high resolution images.

### 4.3 CLASSIFIER AND REGRESSOR PROBES FOR EVALUATION

To study the visual representations of different *frozen* pre-trained models (Section 4.2), our experimental setup involves both classification and regression tasks across different datasets (Section 4.4). In this section, we provide details of task-wise probes.

- **Semantic segmentation.** We use three probes with frozen image encoders: (1) linear head, a point-wise convolutional layer that projects the features of the image encoder to the number of classes in the semantic segmentation dataset. (2) DeepLabv3 head (Chen et al., 2017), a standard non-linear head for dense prediction tasks. (3) PSPNet head (Zhao et al., 2017), another standard and widely used non-linear head for dense prediction tasks. During probing, we minimize the loss between predicted and ground-truth segmentation masks using a cross-entropy loss.

- **Object detection and instance segmentation.** We use two probes with CLIP's frozen image encoder: (1) Mask R-CNN heads (He et al., 2017), a widely used non-linear multi-task head for object detection and instance segmentation. (2) SSD head (Liu et al., 2015), another standard head for efficient object detection. During probing, we minimize with the same classification and localization losses as used in the original works.

- **Monocular depth estimation.** Because it is a dense prediction task, we use the same probes as semantic segmentation task with frozen image encoder. We minimize the scale-shift invariant loss (SSI) Bhat et al. (2023) (in disparity space) between predicted and ground-truth disparity maps.

- **Surface normal estimation.** We use the same probes as segmentation and minimize the the angular loss with learned attenuation (NLL-AngMF) between predicted and ground truth surface normal values (Bae et al., 2021).

- **Image classification.** We use linear (i.e. fully-connected) layer with the frozen image encoder. We minimize cross-entropy loss between predicted and ground-truth labels.

### 4.4 EVALUATION DOWNSTREAM TASKS AND DATASETS

We evaluate different models using classifier and regressor probes (Section 4.3) on the following down-stream tasks:

- **Semantic segmentation.** We use PASCAL VOC (Everingham et al., 2010) with 20 classes and ADE20K (Zhou et al., 2016) with 150 classes for the task of semantic segmentation. Note that, the classes in the PASCAL VOC dataset are a subset of the COCO classes on which the segmentation expert is trained (Section 4.1)). On the other hand, majority of the classes in ADE20k are not part of the segmentation expert's training corpora. Evaluating on these two types of datasets enables us to better understand the true gains of pseudo-supervision from experts. Following a standard convention, we report the accuracy on the validation sets of these datasets in terms of mean intersection over union (mIoU).

- **Object detection and instance segmentation.** We use the COCO dataset for detection and instance segmentation. Importantly, during training with pseudo-labels, we do not use the bounding boxes. Instead, we convert instance masks into semantic segmentation pseudo-labels. This allows us to evaluate baselines on both instance segmentation and object detection, which are considered to be more challenging tasks than semantic segmentation. Following standard convention, we evaluate the accuracy on COCO's validation set in terms of mean average precision (mAP).

- **Monocular depth estimation.** We use NYU-V2 (Nathan Silberman & Fergus, 2012) dataset as our depth estimation benchmark. Note that DPT, the expert used for depth pseudo-supervision, is trained on a different dataset, i.e., ScanNet. We use absolute relative error as a metric for evaluation on the validation set.

- **Surface normal estimation.** We use NYU-V2 for surface normal estimation. We follow Bae et al. (2021); Qi et al. (2018) for training dataset, and evaluate on the official test set of NYU-V2. Importantly, it's worth noting that the surface normal expert (Section 4.1) has been trained on the ScanNet dataset. Therefore, **CLIPTeX** has not been exposed to images and labels from NYU-V2 during the pseudo-supervision process. This setup allows us to ascertain whether **CLIPTeX** can transfer to unseen datasets or not. Following Bae et al. (2021), we use $a{<}30$ as the metric for evaluation.

Table 1: **Probing results for semantic segmentation.** A higher value of mIoU is better.

| Model | ADE20k | | | PascalVOC | | |
|---|---|---|---|---|---|---|
| | **Linear** | **DeepLabV3** | **PSPNet** | **Linear** | **DeepLabV3** | **PSPNet** |
| ViT-B/16 | | | | | | |
| CLIP | 6.78 | 16.15 | 17.32 | 18.66 | 43.75 | 45.53 |
| CLIP-FT | 26.60 | 37.11 | 38.80 | 62.47 | 77.67 | 78.22 |
| **CLIPTeX (Ours)** | **29.26** | **39.20** | **39.70** | **73.43** | **80.57** | **80.71** |
| ViT-H/16 | | | | | | |
| CLIP | 24.18 | 33.39 | 34.86 | 56.18 | 73.12 | 75.37 |
| CLIP-FT | 32.20 | 43.05 | 44.24 | 62.95 | 81.73 | 82.94 |
| **CLIPTeX (Ours)** | **36.17** | **45.43** | **45.63** | **79.30** | **84.06** | **84.31** |
| ResNet-50 | | | | | | |
| CLIP | 11.98 | 29.51 | 28.22 | **46.96** | 70.34 | 70.92 |
| CLIP-FT | 11.30 | 34.86 | 33.97 | 34.78 | 73.70 | 74.17 |
| **CLIPTeX (Ours)** | **12.93** | **35.45** | **34.80** | 40.31 | **75.82** | **75.58** |

- **Image classification.** We evaluate on two standard image classification datasets, i.e., ImageNet (Russakovsky et al., 2014) and Places365 (Zhou et al., 2017). We use top-1 accuracy on the validation set as an evaluation metric.

Details about our implementation, including hyper-parameters, used for training on CC3m and probing tasks are given in Appendix A.

## 5 RESULTS

This section presents the probing results of **CLIPTeX** trained with pseudo-supervision from experts. Our results shows that **CLIPTeX** enhances the visual representations of the image encoder in CLIP, leading to significant improvements across a variety of tasks and datasets when probed using different probes.

### 5.1 PSEUDO-SUPERVISION IMPROVES VISUAL REPRESENTATIONS

**Semantic segmentation.** Probing results for the task of semantic segmentation on PASCAL VOC and ADE20k are given in Table 1. **CLIPTeX** shows consistent improvements over the baselines. Particularly noteworthy is the linear probing accuracy of **CLIPTeX** with ViT-B/16 and ViT-H/16 backbones on the PASCAL VOC dataset, which is about 10% and 16.3% better than CLIP-FT. These substantial improvements can be partially attributed to the fact that the semantic classes in PASCAL VOC are a subset of semantic classes in COCO, on which the segmentation expert is trained. We also observe improvements of up to 2.66% on the ADE20k dataset, despite the fact that the experts were *not* trained on the majority of its classes. These improvements underscores the enhancement in visual representations within the image encoder.

It is worth mentioning that while ViT backbones have generally exhibited superior performance compared to ResNets, we observed that the gap in mIoU between different models with ResNet as image encoders is relatively small in comparison to ViT backbones. This observation is likely attributed to the inductive biases present in CNNs, which ViT-based models may lack. Note that CLIP's mIoU is notably lower in comparison to other models for ViT backbones. This discrepancy is likely attributed to the fact that the CLIP is pre-trained at a resolution of $224 \times 224$, whereas both CLIP-FT and **CLIPTeX** employ a higher resolution of $512 \times 512$.

**Object detection and instance segmentation** Table 2a shows probing results for the task of object detection and instance segmentation. For Mask R-CNN head with ViT-B/16 as the frozen backbone, **CLIPTeX** delivers 13.69% and 1.68% better bounding box mAP over CLIP and CLIP-FT respectively. We observe similar gains when **CLIPTeX** is used with SSD. These results suggest that pseudo-supervision from experts improves CLIP's image representations for localization tasks.

Table 2: **Probing results for object detection, instance segmentation, and image classification.** In (a), for Mask R-CNN, we report mAP (higher is better) for bounding box and instance segmentation while for SSD, we report mAP only for bounding box on the COCO dataset. In (b) top-1 accuracy (higher is better) is reported.

(a) Detection and instance segmentation on COCO.

| Model | Mask R-CNN | | SSD |
|---|---|---|---|
| | **BBox** | **Instance** | **BBox** |
| ViT-B/16 | | | |
| CLIP | 15.20 | 12.16 | 5.33 |
| CLIP-FT | 27.21 | 23.18 | 16.46 |
| **CLIPTeX (Ours)** | **28.89** | **24.92** | **17.50** |
| ViT-H/16 | | | |
| CLIP | 26.65 | 21.29 | 11.07 |
| CLIP-FT | 33.93 | 28.92 | 20.24 |
| **CLIPTeX (Ours)** | **34.50** | **29.60** | **21.55** |
| ResNet-50 | | | |
| CLIP | 29.49 | 25.61 | 20.32 |
| CLIP-FT | 38.13 | 34.02 | **30.28** |
| **CLIPTeX (Ours)** | **38.23** | **34.04** | 28.62 |

(b) Image classification.

| Model | ImageNet | Places365 |
|---|---|---|
| ViT-B/16 | | |
| CLIP | **80.24** | **55.52** |
| CLIP-FT | 79.94 | 55.21 |
| **CLIPTeX (Ours)** | 79.64 | 55.36 |
| ViT-H/16 | | |
| CLIP | **84.85** | **56.96** |
| CLIP-FT | 84.1 | 55.81 |
| **CLIPTeX (Ours)** | 83.2 | 55.96 |
| ResNet-50 | | |
| CLIP | 78.35 | 56.55 |
| CLIP-FT | 78.92 | 56.98 |
| **CLIPTeX (Ours)** | **78.95** | **57.22** |

Table 3: **Probing results for depth and surface normal estimation on NYU-V2 dataset.** Following Lasinger et al. (2019), we report absolute relative error (lower is better) for depth estimation. For surface normal estimation, we report $a<30$ following Bae et al. (2021) (higher is better).

(a) Depth estimation.

| Model | Linear | DeepLabV3 | PSPNet |
|---|---|---|---|
| ViT-B/16 | | | |
| CLIP | 0.235 | 0.189 | 0.168 |
| CLIP-FT | 0.215 | 0.145 | 0.139 |
| **CLIPTeX (Ours)** | **0.159** | **0.129** | **0.128** |
| ViT-H/16 | | | |
| CLIP | 0.212 | 0.151 | 0.132 |
| CLIP-FT | 0.213 | 0.131 | 0.125 |
| **CLIPTeX (Ours)** | **0.138** | **0.118** | **0.117** |
| ResNet-50 | | | |
| CLIP | **0.212** | 0.156 | **0.147** |
| CLIP-FT | 0.239 | 0.160 | 0.155 |
| **CLIPTeX (Ours)** | 0.220 | **0.153** | 0.150 |

(b) Surface normal estimation.

| Model | Linear | DeepLabV3 | PSPNet |
|---|---|---|---|
| ViT-B/16 | | | |
| CLIP | 28.49 | 45.17 | 47.29 |
| CLIP-FT | 29.06 | 47.74 | 47.91 |
| **CLIPTeX (Ours)** | **39.96** | **50.95** | **50.80** |
| ViT-H/16 | | | |
| CLIP | 29.09 | 47.31 | 49.78 |
| CLIP-FT | 29.21 | 49.73 | 50.48 |
| **CLIPTeX (Ours)** | **43.22** | **53.23** | **53.89** |
| ResNet-50 | | | |
| CLIP | **33.67** | 46.05 | 47.28 |
| CLIP-FT | 28.72 | 46.99 | 48.66 |
| **CLIPTeX (Ours)** | 31.56 | **47.92** | **49.44** |

Interestingly, ResNet-50-based CLIP models with different detection heads delivered better accuracy than ViT-based models. Our findings align with that of ViT-Det (Li et al., 2022b) which also indicates that ResNet-based models with less capacity delivers similar or better performance than ViT-based models on transfer learning for object detection.

**Depth and surface normal estimation.** `CLIPTeX` obtains lower error rate (Table 3a) and higher value of $a<30$ (Table 3b) as compared to CLIP and CLIP-FT across various probing heads on NYU-V2 for depth estimation and surface normal estimation respectively. These results indicate a positive transfer of distance and surface normal knowledge to `CLIPTeX`'s image backbone, contributing to the improved performance.

**Image classification.** Unlike other dense prediction tasks discussed above, we observe that CLIP achieves similar or slightly better accuracy compared to CLIP-FT and `CLIPTeX` (see Table 2b). This outcome can be attributed to the characteristics of image classification tasks, which typically involve assigning a single label to an entire image based on its 2D visual content. These tasks primarily focus on recognizing objects without requiring detailed information about object boundaries, spatial relationships, or the 3D structure of the scene.

Table 4: **CLIP's zero-shot knowledge is preserved when trained with experts.** Following CLIP (Radford et al., 2021), we report zero-shot top-1 accuracy for ImageNet-1k dataset and recall@1/5/10 for Flickr-30k dataset.

(a) 0-shot classification on ImageNet.

| Model | 0-shot Top-1 |
|---|---|
| CLIP-FT | 68.76 |
| **CLIPTeX (Ours)** | 68.25 |

(b) 0-shot retrieval on Flickr-30k.

| Model | Text Retrieval | | | Image Retrieval | | |
|---|---|---|---|---|---|---|
| | R@1 | R@5 | R@10 | R@1 | R@5 | R@10 |
| CLIP-FT | 85.90 | 96.70 | 98.60 | 71.66 | 91.00 | 94.94 |
| **CLIPTeX (Ours)** | 86.00 | 96.90 | 98.70 | 71.40 | 90.86 | 95.16 |

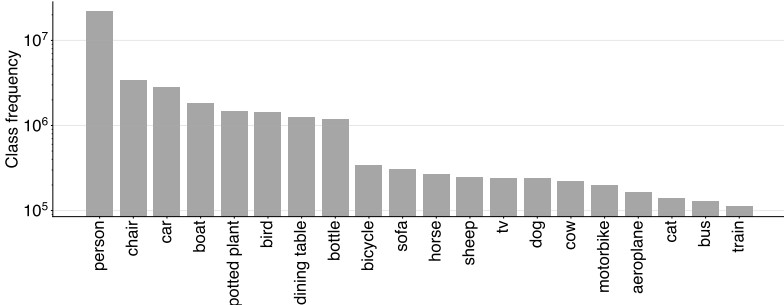

(a) Bounding box frequency for PASCAL VOC classes in CC3M's pseudo-labels obtained with Mask R-CNN.

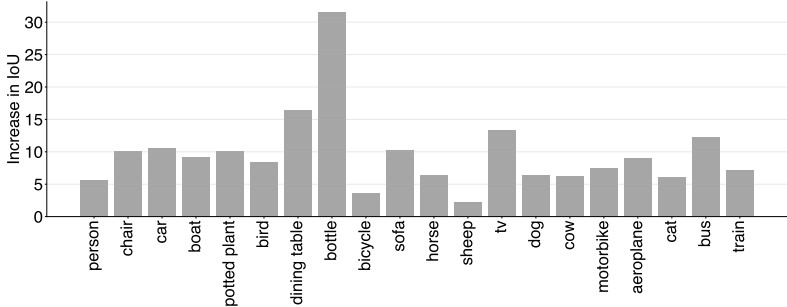

(b) Class-wise IoU gap (in %) between CLIP-FT and **CLIPTeX** when linear probed on the PASCAL VOC.

Figure 2: **Positive transfer with CLIPTeX.**

## 5.2 ZERO-SHOT CAPABILITIES ARE PRESERVED IN CLIPTeX

One of the important and powerful characteristics of CLIP is *prompting*, which enables zero-shot transfer to new datasets. Pseudo-supervision with experts can potentially lead to catastrophic forgetting of previously learned knowledge, which may in turn affect the model's zero-shot generalization capabilities. Table 4 compares the zero-shot capabilities of CLIP-FT and **CLIPTeX** in classification on ImageNet-1k (Russakovsky et al., 2014) and retrieval on Flickr-30k (Young et al., 2014) tasks respectively. **CLIPTeX**'s zero-shot performance is on par with that of CLIP-FT, indicating that enhanced representations do not result in catastrophic forgetting.

## 5.3 POSITIVE TRANSFER OF REPRESENTATIONS FROM CLIPTeX TO DOWNSTREAM TASKS

The CC3M dataset is uncurated and noisy, and may have a skewed distribution towards specific object classes or scenes. Consequently, knowledge transfer from experts to **CLIPTeX** may also be skewed towards more frequent objects in the data. To explore this phenomenon, we quantified the frequency of objects (bounding boxes or instances) in the pseudo-labels generated by the Mask R-CNN expert (Fig. 2a) on the CC3M dataset. Additionally, we examined class-wise improvements in IoU of **CLIPTeX** with respect to CLIP-FT on the PASCAL VOC dataset (Fig. 2b). **CLIPTeX** improves the IoU for all classes in the PASCAL VOC dataset and is not biased towards the most

Table 5: **Role of pseudo-labels from experts in `CLIPTeX` training.** Results with different probes for different dense prediction tasks are reported (see Section 4 for details). For segmentation, we report the results on the PASCAL VOC dataset. We observe similar trends on ADE20k dataset.

| Row | Expert | | | Segmentation (↑) | | Detection (↑) | | Depth (↓) | | Surface Normal (↑) | |
|---|---|---|---|---|---|---|---|---|---|---|---|
| # | Segmentation | Depth | Surface Normal | Linear | PSPNet | Mask R-CNN | SSD | Linear | PSPNet | Linear | PSPNet |
| R1 | ✗ | ✗ | ✗ | 62.47 | 78.22 | 27.21 | 16.46 | 0.215 | 0.139 | 29.06 | 47.91 |
| R2 | ✓ | ✗ | ✗ | 72.21 | 81.39 | 28.54 | 17.58 | 0.203 | 0.136 | 34.86 | 48.62 |
| R3 | ✗ | ✓ | ✗ | 64.50 | 81.16 | 27.75 | 16.70 | 0.170 | 0.131 | 35.21 | 49.51 |
| R4 | ✗ | ✗ | ✓ | 63.28 | 81.48 | 27.69 | 16.81 | 0.193 | 0.134 | 37.42 | 50.71 |
| R5 | ✓ | ✓ | ✗ | 73.96 | 81.49 | 28.83 | 17.57 | 0.162 | 0.130 | 37.05 | 49.69 |
| R6 | ✓ | ✗ | ✓ | 72.67 | 81.30 | 28.83 | 17.75 | 0.188 | 0.132 | 38.65 | 50.48 |
| R7 | ✗ | ✓ | ✓ | 64.20 | 81.17 | 27.90 | 17.00 | 0.165 | 0.129 | 39.59 | 51.01 |
| R8 | ✓ | ✓ | ✓ | 73.43 | 80.71 | 28.89 | 17.50 | 0.159 | 0.128 | 39.96 | 50.49 |

Table 6: **Role of head complexity (light and heavy) when training with pseudo-labels on CC3m.** #layers denote the number of convolutional layers used in the task head. Results with different probes for different dense prediction tasks are reported (see Section 4 for details). For segmentation, we report the results on the PASCAL VOC dataset. We observe similar trends in ADE20k dataset.

| # layers | Segmentation (↑) | | Detection (↑) | | Depth (↓) | | Surface Normal (↑) | |
|---|---|---|---|---|---|---|---|---|
| | Linear | PSPNet | Mask R-CNN | SSD | Linear | PSPNet | Linear | PSPNet |
| 1 | 73.43 | 80.71 | 28.89 | 17.50 | 0.159 | 0.128 | 39.96 | 50.80 |
| 3 | 66.70 | 80.24 | 28.64 | 17.43 | 0.155 | 0.127 | 40.55 | 51.72 |

frequently occurring object classes. These findings, combined with insights in Section 5.1 suggests positive transfer of representations from `CLIPTeX` to down-stream tasks.

### 5.4 ABLATIONS

**Role of pseudo-labels from experts in training `CLIPTeX`.** Incorporating pseudo-supervision from task-specific experts, even from a single expert during training, results in substantial improvements in performance. These improvements are observed when evaluating models on various downstream tasks with different probes (see R1 vs. rest; Table 5). Overall, our findings indicate that incorporating knowledge from all experts contributes to learning better visual representations. Therefore, we use all experts for pseudo-supervision while training `CLIPTeX`.

**Task-head complexity.** As discussed in Section 3, we use light-weight heads to improve visual representations in CLIP's image encoder. We replace these heads with heavier counterparts (comprising of three standard convolutional layers) when training `CLIPTeX` with CC3M pseudo-labels. Table 6 shows that light-weight heads deliver similar performance to heavy-weight heads in most cases. Therefore, we use light-weight heads for pseudo-supervision in our experiments to make the training more efficient.

### 6 CONCLUSION

As the field of machine learning research embraces openness, a growing number of specialized expert models become publicly available. Our study showcased the potential of leveraging these publicly available expert models to enhance CLIP's visual representations, all without the necessity of collecting task-specific data. Our experiments revealed that `CLIPTeX` yields improvements across a wide range of tasks, highlighting its versatility and effectiveness.

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

## A  HYPERPARAMETERS

Hyper-parameters used during training and probing `CLIPTeX` and other models are given in Table 7 and Table 8 respectively.

Table 7: **Hyper-parameters for training `CLIPTeX` on CC3M dataset..**

| Hyper-parameter | Value |
|---|---|
| Epochs | 30 |
| LR scheduler | cosine |
| Warmup Steps | 1000 |
| Warmup Init LR | 1e-06 |
| Maximum LR | 3e-05 |
| Minimum LR | 1e-06 |
| Batch size | 32 |
| $\lambda_{\text{depth}}$ | 1.0 |
| $\lambda_{\text{clip}}$ | 1.0 |
| $\lambda_{\text{seg}}$ | 0.1 |
| $\lambda_{\text{surface normal}}$ | 1.0 |

Table 8: Hyper-parameters used for probing on different downstream tasks.

| Hyper-paramater | Segmentation | | | Detection | | Depth | | | Surface Normal | | | Classification |
|---|---|---|---|---|---|---|---|---|---|---|---|---|
| | Linear | DeepLabv3 | PSPNet | Mask R-CNN | SSD | Linear | DeepLabv3 | PSPNet | Linear | DeepLabv3 | PSPNet | Linear |
| Epochs | 50 | 50 | 50 | 25 | 200 | 50 | 50 | 50 | 50 | 50 | 50 | 40 |
| LR scheduler | cosine | cosine | cosine | multi-step | cosine | cosine | cosine | cosine | cosine | cosine | cosine | cosine |
| Warmup Steps | 500 | 500 | 500 | 250 | 500 | 1000 | 1000 | 1000 | 1000 | 1000 | 1000 | 1000 |
| Warmup Init LR | 1e-06 | 1e-06 | 1e-06 | 1e-05 | 9e-05 | 1e-06 | 1e-06 | 1e-06 | 1e-06 | 1e-06 | 1e-06 | 1e-06 |
| Maximum LR | 3e-05 | 3e-05 | 3e-05 | 3e-04 | 9e-04 | 1e-04 | 1e-04 | 1e-04 | 1e-05 | 1e-05 | 1e-05 | 3e-05 |
| Minimum LR | 3e-06 | 3e-06 | 3e-06 | NA | 1e-06 | 1e-06 | 1e-06 | 1e-06 | 1e-06 | 1e-06 | 1e-06 | 1e-06 |
| LR Milestones | NA | NA | NA | [22, 24] | NA | NA | NA | NA | NA | NA | NA | NA |
| LR Gamma | NA | NA | NA | 0.1 | NA | NA | NA | NA | NA | NA | NA | NA |
| Batch size | 32 | 32 | 32 | 4 | 32 | 16 | 16 | 16 | 16 | 16 | 16 | 128 |

