# OpenReview forum: "CLIP meets Model Zoo Experts: Pseudo-Supervision for Visual Enhancement"
_ICLR.cc/2024/Conference — ICLR 2024 Conference Withdrawn Submission_

### Official Review · Reviewer_BcRN · 2023-10-29

**Soundness:** 3 good
**Presentation:** 3 good
**Contribution:** 2 fair
**Rating:** 3
**Confidence:** 4

**Summary:**

This paper proposes a training method to improve the CLIP’s visual representation based on task-specific vision models. It utilizes the vision models from model zoo to construct pseudo labels for noisy image-text models, serving as extra supervision besides the contrastive loss. This simple method is effective, improving  up to 16.3% across different vision tasks, including segmentation, detection, depth estimation, and surface normal estimation.

**Strengths:**

1. The proposed method is simple yet effective, leveraging existing vision models to serve as teacher for extra supervision. The improvements is obvious even compared to fine-tuned CLIP model on CC3M dataset.
2. The effectiveness is demonstrated on a bunch of downstream tasks, including segmentation, detection, depth estimation, and surface normal estimation across multiple datasets.

**Weaknesses:**

Limitations of novelty.  The paper claims proposed method uses publicly accessible experts trained on diverse tasks with different data distributions and objectives, which is different from previous works that use vision foundation models to generate labels. However, from the Fig.1 and model design, data samples are labeled by various foundation models and losses are computed respectively to optimize task heads, which is similar to previous pseudo labeling strategy.

**Questions:**

The training process involves multiple vision foundation model forwarding process, which would slowen the training process. How much impact will this have on the training process? And is it fair to compare the training strategy with CLIP-FT model in paper?

---

### Official Review · Reviewer_hJxN · 2023-10-30

**Soundness:** 2 fair
**Presentation:** 3 good
**Contribution:** 2 fair
**Rating:** 3
**Confidence:** 5

**Summary:**

This paper aims to augment CLIP training with task-specific data and task heads. In particular, the authors use open-source task-specific vision models to generate the pseudo-labels and train the task-specific heads using these labels. The experiment results show the effectiveness of training such CLIP model while keeping zero-shot classification ability.

**Strengths:**

- Well written and easy to follow.

- The motivation is clear and idea is simple to understand.

- The experiment results show the effectiveness of pseudo-label training in different tasks, including segmentation, detection, and depth estimation.

**Weaknesses:**

- The experiment results are not convincing. The baselines are not strong. The authors should present more strong baselines, including Mask2Former. Moreover, this work dose not compare with recent state-of-the-art approach whether on semantic segmentation or depth prediction.

- Missing the frozen trained CLIP model baselines with heavier head [1], [2], [3]. What are the Frozen CLIP results of strong baselines?

- The ablation studies are not good. For example, the effects of various task heads are not explored. The effects of different task-specific experts are not explored.
The experiment details can be put into appendix.
- In abstract, “it lacks object localization capabilities” Personally, CLIP models have the localization ability. Several works [1][2] have adopted CLIP as feature extractor, which also achieve good results.

- Figure-1 (c) needs to add the baseline results for better comparison.


[1], Frozen clip models are efficient video learners, ECCV-2022

[2], Convolutions Die Hard: Open-Vocabulary Segmentation with Single Frozen Convolutional CLIP, NeurIPS-2023
[3]. F-VLM: Open-Vocabulary Object Detection upon Frozen Vision and Language Models, ICLR-2023

**Questions:**

See the weakness part.

---

### Official Review · Reviewer_8Cdu · 2023-11-01

**Soundness:** 2 fair
**Presentation:** 3 good
**Contribution:** 1 poor
**Rating:** 3
**Confidence:** 5

**Summary:**

This paper proposes CLIPTeX, which enhances CLIP's capabilities utilizing specialized vision models.
By generating pseudo-labels from these models and subsequently training CLIP on these labels combined with image-text pairs, the approach has shown notable improvements in various vision tasks.

CLIPTeX not only bolsters CLIP's visual understanding but also preserves its foundational strengths, ensuring its applicability across several computer vision tasks. This paper conducts experiments across multiple datasets to demonstrate the potential of CLIPTeX.

**Strengths:**

1. This paper is well-written and easy to follow.
2. The rigorous experimentation across diverse tasks such as segmentation, detection, depth estimation, and surface normal estimation lends credibility to the paper's claims.
3. This work emphasizes the potential of using pseudo-labels, setting a precedent for future research to consider such augmentation strategies.

**Weaknesses:**

1. The pre-processing to get the pseudo label is somehow time-consuming.
2. Considering CLIP is a vision-language pre-training model, evaluation results on the cross-modal downstream tasks are necessary, which helps demonstrate the cross-modal dense understanding capability of proposed CLIPTeX, such as 2D visual grounding, 2D question-answering, etc.
3. The reviewer holds that the novelty of this paper is limited. Instead of introducing a fundamentally new approach or technique, the paper's main contribution is in integrating specialized task-specific vision models with CLIP. While this integration does lead to performance improvements, the core idea revolves around a simple application of pseudo-labels. Essentially, the work can be viewed as a refinement of CLIP without enough novelty.
4. Besides quantitative results, qualitative results on downstream tasks are required to further prove the 2D representation capability of CLIPTeX.

**Questions:**

Please check the Weaknesses mentioned above.

---

### Official Review · Reviewer_Q843 · 2023-11-08

**Soundness:** 3 good
**Presentation:** 3 good
**Contribution:** 3 good
**Rating:** 8
**Confidence:** 3

**Summary:**

In this work, the authors augment the capabilities of CLIP with task-specific experts that help to improve its representation for the downstream tasks. Those experts are well-known models from model zoos used to create hard pseudo-labels on web-scale noisy image-text datasets.

**Strengths:**

- Easy to read. Good experiments and ablation.
- It is great to see that by using experts and doing contrastive task-specific loss, the performance on downstream task improve, and CLIP maintains its versatility and obtain comparable performance on zero-shot classification
- The method is simple and efficient.

**Weaknesses:**

- It is interesting to see that the complementary task help between each others. Table 5, I believe lot of insights can be done and I was expecting to see more analysis in this part of the paper.
- It would be great to guess what set of tasks should be pick, for the downstream task. So, we can get a set of different CLIPTeX trained with the combinatories of task/experts so people can use the one that is more likely to work for the downstream task.
For example, for segmentation seems to be that the most valuable experts are the segmentation and depth for linear and PSPNet. Similar to SSD in detection. etc...

**Questions:**

- What is the proof that CLIP is more robust to dataset distribution shifts? Reference, experiments?
- Why Mask R-CNN needs LR milestones and gamma?